# One-Hot Encoding Strikes Back: Fully Orthogonal Coordinate-Aligned Class Representations

## Abstract

Representation learning via embeddings has become a central component in many machine learning tasks. This featurization process has gotten gradually less interpretable from each coordinating having a specific meaning (e.g., one-hot encodings) to learned distributed representations where meaning is entangled across all coordinates. In this paper, we provide a new mechanism that converts state-of-the-art embedded representations and carefully augments them to allocate some of the coordinates for specific meaning. We focus on applications in multi-class image processing applications, where our method Iterative Class Rectification (ICR) makes the representation of each class completely orthogonal, and then changes the basis to be on coordinate axes. This allows these representations to regain their long-lost interpretability, and demonstrating that classification accuracy is about the same or in some cases slightly improved.

## 1 Introduction

Embedded vector representations of structured data objects are nowadays a common intermediate goal for much of machine learning. The goal of these representations is typically to transform data into a form that is easy to work with for downstream applications, most centrally classification tasks. If the representations are successful, then for direct tasks, only a simple classifier is required afterward, e.g., logistic regression.

In this work, we argue that due to the fundamental nature of these representations, they should also aim for explicit interpretability. Note that this is not attempting to make the process or neural architecture parameters used in computing these representations interpretable but that given a data point's vector structure, one should understand the components of its representation. In particular, we argue that for labeled classes provided among training data, we should be able to (a) associate these classes with class mean vectors, (b) these class mean vectors should be completely orthogonal, and (c) each should be aligned with a particular coordinate (a one-hot encoding).

Given such an embedding of data points, then many tasks can be done directly by simply reading the representation. A multi-class classification task can be solved by returning the class associated with the coordinate with the largest value. To understand a data point's relative association among multiple classes, one can compare their coordinate values; note that there are no hidden associations due to full orthogonality. If one fears there is implicit bias in a task, and that bias is associated with a captured class (e.g., gender bias captured by "woman" or "man" class), one can remove that class via projection like in Bolukbasi et al. (2016); Dev & Phillips (2019) – by simply not using those coordinates in downstream analysis. Other tasks without association with the bias should be unaffected, while those contaminated with bias will have that component removed.

A couple of recent papers have attempted to use neural networks to learn embedded representations that have class means orthogonal – their goal was increased generalization. The orthogonal projection loss (OPL) (Ranasinghe et al., 2021), and CIDER (Ming et al., 2023) both add a regularization term which favors compactness among points within a class and near orthogonality among class means. While these methods are useful seeding for our approach, we observe that they fail to produce class means that are nearly

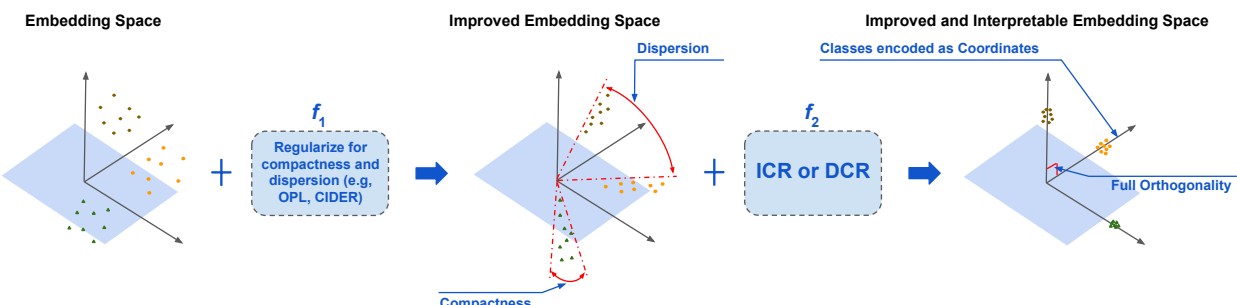

Figure 1: Our approach for embedding multi-class data: $f_1$ initializes classes to be clustered and dispersed. In $f_2$, our ICR and DCR make classes completely orthogonal along the coordinate axis.

orthogonal. The average dot-product between normalized class means on CIFAR-100 for these prior methods is about 0.2; for ours, it is below 0.01.

Furthermore, our proposed framework structurally restricts the classifier to *classification-by-nearest-mean*, also known as the Rocchio algorithm. This directly reflects the training data: for each class, the mean of the training data is stored, and on evaluation of a test point, it is assigned a label of the nearest mean vector. This classification model produces a linear classifier with only two (2) classes, and its standard evaluation reduces to the common task in information retrieval. This multi-class classifier becomes especially effective when the representation of the data is learned and is common among state-of-the-art models (Yu et al., 2020) for few-shot learning approaches in image processing.

Our paper achieves the following:

1. We propose two new class rectification methods (ICR and DCR) for multi-class classification under the Rocchio algorithm, which completely orthogonalize class means.

2. We prove that these methods either require one step (DCR) or iteratively converge to an orthogonal representation (ICR), conditioned that the class data is already clustered.

3. We show that this orthogonalized representation maintains state-of-the-art performance in a variety of classification tasks, given a backbone architecture.

The iterative class rectification (ICR) at the heart of our approach is an extension of a recent method ISR (Aboagye et al., 2023) designed for bias reduction in natural language. That approach, ISR, requires subspaces defined by two opposing classes (e.g., male-female, pleasant-unpleasant), which is restrictive. That paper only found a handful of such classes with sufficient training data, demonstrated the approach converged with two subspaces (2 pairs of classes) and did not always quickly converge to orthogonal on three subspaces (3 pairs of classes). A challenge addressed in that paper was determining a proper point of rotation. By using single-class means, as we propose, this challenge goes away, and we show our approach effortlessly scales to 100 classes. We also introduce a second-class rectification method (DCR), which achieves this result without iteration but has less continuity in how it augments the representation space.

After class means are fully orthogonal, we align them to coordinate axes. This basis transformation, by an orthogonal matrix, does not change any distance or dot-products between data representations.

**Example Uses.** Consider the CIFAR-100 test image with the label orange; see Figure 2.The largest dot-products among the normalized class mean vectors for our technique (OPL+)ICR is with orange (0.995), the correct class, and then a big drop to cockroach at 0.0087 and other smaller values. In contrast, the normalized class mean vectors for other approaches still identify orange as the correct class but have a much larger association with other classes. For OPL, it is orange at 0.9975, but also apple, pear, and sweet_pepper between 0.82 and 0.72. Since the image is so associated with the class mean (dot product of virtually 1), we ascertain that the issue is the class means are not sufficiently orthogonal so that the image earns spurious

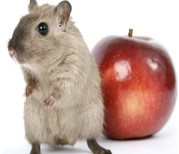

| | | | | | |
|---|---|---|---|---|---|
| top dot products for OPL | 0.9975 orange | 0.8204 apple | 0.7880 pear | 0.7215 sweet_pepper | 0.4562 poppy |
| top dot products for OPL+ICR | 0.9950 orange | 0.0087 cockroach | 0.0061 maple_tree | 0.0059 girl | 0.0051 orchid |

Figure 2: Dot Products with class mean vectors for **orange** image with OPL and OPL+ICR.

| | | | | | | |
|---|---|---|---|---|---|---|
| top dot products for OPL | 0.8832 hamster | 0.7938 rabbit | 0.7681 mouse | 0.7168 squirrel | 0.6873 possum | 0.6266 fox |
| top dot products for OPL+ICR | 0.7621 hamster | 0.4148 apple | 0.2396 pear | 0.2030 squirrel | 0.1880 kangaroo | 0.1331 baby |

Figure 3: Dot products with class mean vectors for **hamster+apple** image with OPL and OPL+ICR.

correlation with the other classes. However, with ICR, this does not happen since the class means are forced to be orthogonal.

Next, in Figure 3 we consider an image that has two classes present: hamster and apple. For OPL+ICR, this image's representational vector has the largest dot-products, with the normalized class means of 0.76 for hamster and 0.41 for apple. The next largest drops to 0.24 for pear and 0.20 for squirrel. In contrast, for OPL, the largest dot products are 0.88 for hamsters, but then the next largest are for rabbits, mice, squirrels, possums, and foxes, all between 0.63 and 0.79. Because the hamster class has a correlation with the other small fury mammals under OPL, they obscure the association with the hamster and hide the association with the apple, which has a score of 0.58. This is not the case with OPL+ICR, so the association with pear and squirrel can be interpreted to geometrically represent uncertainty about those class labels.

Then, we can consider removing the "hamster" class via a projection-based approach (e.g., Dev & Phillips (2019)). Under OPL+ICR, the largest dot-product is now apple, still at 0.41, and the next largest is unchanged with pear (0.24) and squirrel (0.20). For OPL after projection, we also have the largest dot-product with apple at 0.41, but somewhat obscured with other correlated classes, including pear, orange, and sweet_pepper, all between 0.33 and 0.38. Notably, the other small fury mammals are also removed from strong association because of their correlation with the hamster class.

## 2 Algorithmic Framework

Our method considers a data set $Z \subset \mathcal{Z}$, where each $z_i \in Z$ is associated with a label $y_i \in [k]$, where $k$ is the number of distinct classes. We use image data $\mathcal{Z}$ as an exemplar. Then, it operates in two phases towards creating an embedding in $\mathbb{R}^d$, with $d > k$; see Figure 1. The first phase learns an embedding $f_1 : \mathcal{Z} \to \mathbb{R}^d$ with the goal of classes being (nearly) linearly separable in $\mathbb{R}^d$. The second phase, the innovation of this paper, is another map $f_2 : \mathbb{R}^d \to \mathbb{R}^d$, which aims to retain (and perhaps improve) linear separability but also achieve a form of orthogonality among classes. While this second phase can be interpreted as a form of learning–so it only sees training and not testing data–it is deterministic and does not follow the traditional *optimize parameters over a loss function*.

For input data $(Z, y)$, denote $X' = \{x'_i = f_1(z_i) \in \mathbb{R}^d \mid z_i \in Z\}$ as the embedding after phase 1. Then denote $X = \{x_i = f_2(x'_i) \in \mathbb{R}^d \mid x'_i \in X'\}$ as the embedding after phase 2. Let $Z_j$, $X'_j$, and $X_j$ be the data points in class $j \in [k]$ for the initial data, first, and final embedding, respectively.

**Rocchio algorithm.** We leverage the Rocchio algorithm to build classifiers. For an embedded data set $(X, y)$, it first creates class means $v_j = \frac{1}{|X_j|} \sum_{x_i \in X_j} x_i$ for each class $j \in [k]$. Then on a training data point $x \in \mathbb{R}^d$ it predicts class $\hat{j} = \arg\min_{j \in [k]} \mathrm{D}(x, v_j)$. If we normalize all class means (so $v_j \leftarrow v_j / \|v_j\|$)

then using Euclidean $\text{D}(x, v_j) = \|x - v_j\|$ has the same ordering as cosine distance. Instead, we can use $\hat{j} = \arg\max_{j \in [k]} \langle x, v_j \rangle$; we do this hereafter unless stated otherwise.

**Phase 1 embedding.** For the first phase embeddings $f_1$, we leverage existing recent algorithms that aim for an embedding with three goals: (a) *accuracy*: each class can be (nearly) linearly separable from all other classes. (b) *compactness*: each class $X'_j$ has points close to each other, i.e., small variance. (c) *dispersion*: each pair of classes $j$ and $j'$ are separated, and in fact nearly orthogonal. In particular, a couple of recent papers proposed loss functions for $f_1$ as $\mathcal{L}_{f_1} = \mathcal{L}_{CE} + \lambda(\mathcal{L}_{comp} + \mathcal{L}_{disp})$. The $\mathcal{L}_{CE}$ is the standard cross entropy loss which optimizes (a), $\lambda \in [0, 1]$, and where $\mathcal{L}_{comp}$ and $\mathcal{L}_{disp}$ optimize (b) and (c). These are actualized with $|Z| = n$, $k$ classes, $n_1 = \sum_{j \in [k]} |Z_j|(|Z_j| - 1)$ and $n_2 = \sum_{j \in [k]} |Z_j|(n - |Z_j|)$ as:

$$\mathcal{L}_{comp} = 1 - \frac{1}{n_1} \sum_{j \in [k]} \sum_{z_i, z_{i'} \in Z_j} \langle f_1(z_i), f_1(z_{i'}) \rangle, \qquad \mathcal{L}_{disp} = \left| \frac{1}{n_2} \sum_{z_i \in Z_j; z_{i'} \in Z_{j' \neq j}} \langle f_1(z_i), f_1(z_{i'}) \rangle \right| \qquad (1)$$

$$\mathcal{L}_{comp} = -\frac{1}{n} \sum_{i=1}^{n} \log \frac{\exp(\langle f_1(z_i), v_{j_i} \rangle)}{\sum_{j=1}^{k} \exp(\langle f_1(z_i), v_j \rangle)}, \qquad \mathcal{L}_{disp} = \frac{1}{k} \sum_{j \in [k]} \log \frac{1}{k-1} \sum_{j' \neq j} \exp(\langle v_j, v_{j'} \rangle) \qquad (2)$$

The loss for OPL (Ranasinghe et al., 2021) is in eq 1 and for CIDER (Ming et al., 2023) in eq 2.

We observe (see Table 1) that these achieve good clustering among classes, but the classes are not fully orthogonal. On training data for CIFAR-100, they achieve about 98% accuracy or better. This holds under the trained linear classifier (under logistic regression) or the Rocchio algorithm. Pre-processing in phase 1 will prove an important first step for the success of our phase 2.

**Phase 2 embedding: Full orthogonalization.** As we observe that the result of *learning* an orthogonal embedding through regularization is not completely effective, the second phase provides a deterministic approach that *enforces* orthogonality of the class means. A first – but unworkable – thought is to *just run Gram-Schmidt* on the class mean vectors $v_1, \ldots, v_k$. However, this does not produce a generic function $f_2$ that also applies to training data; we want to transform $X$ by some $f_2$, so if we recalculate their class means under $f_2(X)$, they are then orthogonal. But Gram-Schmidt only explains what to do for vectors $v_1, \ldots, v_k$, not a general function $f_2$ that applies to other embedded data $X$. Toward this end, we propose two approaches: ICR and DCR.

Iterative Class Rectification (ICR): For ICR, we adapt a recent approach called Iterative Subspace Rectification (ISR) (Aboagye et al., 2023) designed to orthogonalize language subspaces to reduce bias. This approach handles two concepts, each defined by a pair of classes (e.g., male-female, pleasant-unpleasant) as vectors $v_1, v_2$, and centers the data around these. Then it applies a "graded rotation" operation (Dev et al., 2021) (see Algorithm 5 in the Appendix) to the components of each $x \in X$ that lies within the span of the two linear concept directions: $\text{span}(v_1, v_2)$. Because it operates only in this span, it only alters the embedding of each $x \in X$ in this 2-dimensional span, denoted $\pi_{\text{span}(v_1, v_2)}(x)$ and shortened to $\pi(x)$ in the algorithms. The graded rotation is designed so the operation when defined on $v_1$ and $v_2$, is easy to understand: it moves $v_2 \mapsto v'_2$ so it is orthogonal to $v_1$, and it does not change $v_1$. For every other $x' = \pi_{\text{span}(v_1, v_2)}(x)$, the graded rotation defines a rotation that depends on the angle to $v_1$. Those closer in angle to $v_1$ rotate very little, and those closer in angle to $v_2$ rotate an angle almost as much as $v_2 \mapsto v'_2$. The magnitude of this angle varies continually based on the angle from $v_1$. The full technical details are explained in Dev et al. (2021) and are reproduced in full in Algorithm 5 in the Appendix. The ISR paper (Aboagye et al., 2023) demonstrates empirically that by *repeating* this process we get $v_2 \mapsto v_2^\star$, with $v_2^\star$ orthogonal to $v_1$, *even recomputing $v_1$ and $v_2^\star$* from the updated embedded data points which define the associated classes.

**Algorithm 1** BinaryICR($X, X_1, X_2$, iters: $T$)

---
1: **for** $i = 0, 1, \ldots, T-1$ **do**
2:     $v_1, v_2 \leftarrow$ normalized means($X_1, X_2$)
3:     BinaryCR($X, v_1, v_2$)

---

**Algorithm 2** BinaryCR($X, u, v$)

---
1: Set $v' = v - \langle u, v \rangle u$
2: Define projection $\pi(\cdot) = (\langle \cdot, u \rangle, \langle \cdot, v' \rangle)$
3: **for** $x \in X$ **do**
4:     $\tilde{x} \leftarrow \text{GradedRotat}(\pi(u), \pi(v), \pi(x))$
5:     $x \leftarrow x + (\langle \pi(u), \tilde{x} - \pi(x) \rangle u + \langle \pi(v'), \tilde{x} - \pi(x) \rangle v')$

---

We adapt this process in two ways in this paper, in Algorithms 1 and 2. First, we only use individual classes and their class means (line 2 of Alg 1) in place of concepts that spanned across two opposing ideas (and hence two sets of embedded points for each concept). Second, because we initialize with clustered concepts *by cosine similarity* around their class mean vectors, we can rotate around the origin (line 4 of Alg 2) and do not require a centering step as in ISR. Algorithm 2 does the core operation of projecting to the span of two subspaces $u, v$, apply GradedRotation on each point $x \in X$ and then adjust only the coordinates in span($u, v$) (line 5). Algorithm 1 iterates this procedure $T$ steps as the recalculated class means become orthogonal.

**Algorithm 3** MultiICR($X, X_1, \ldots, X_k$, iters: $T$)

---
1: **for** $i = 0, 1, \ldots, T-1$ **do**
2:     Let $v_i$ be the normalized mean vector of class $X_i$ for $i = 1, 2, \ldots, k$.
3:     Set $r, s = \arg\min_{1 \leq i,j \leq k} |\langle v_i, v_j \rangle|$, WLOG suppose $r = 1$, $s = 2$
4:     Let $S_1$ and $S_2$ be the span of $\{v_1\}$ and $\{v_1, v_2\}$ respectively
5:     Run BinaryCR($X, v_1, v_2$)
6:     Recalculate normalized class means $v_i$ for all $i$
7:     **for** $i = 3, \ldots, k$ **do**
8:       Choose $t = \arg\min_{j \geq i} \langle v_1, v_j \rangle^2 + \langle v_2, v_j \rangle^2 + \cdots \langle v_{i-1}, v_j \rangle^2$
9:       WLOG assume $t = i$
10:      Let $\bar{v}_i$ be the projection of $v_i$ onto $S_{i-1}$
11:      Set $u_i = v_i - \sum_{j=1}^{i-1} \langle v_j, v_i \rangle v_j$ and $v_i' = u_i / \|u_i\|$
12:      Run BinaryCR($X, v_i', \bar{v}_i$)
13:      Set $S_i$ to be the span of $\{S_{i-1}, v_i\}$
14:      Recalculate class normalized means $v_j$ for all $j$

---

To apply this to all classes, we now apply a Gram-Schmidt sort of procedure; see details in Algorithm 3. We first identify the class mean vectors that are most orthogonal (line 3) and apply one step of Binary ICR. Then, at each round, we find and maintain the subspace of the class means we have attended to so far $S_{j-1}$, and find the class mean $v_j$ most orthogonal to that subspace (line 8). We project $v_j$ onto $S_{j-1}$, to get $\bar{v}_j$, and then run one step of BinaryCR to orthogonalize $v_j$ from $\bar{v}_j$ (and hence to all of $S_{j-1}$). Once we have addressed all classes, we iterate this entire procedure a few times (typically $T = 1$ or 2 iterations, and not more than 5).

Finally, at the conclusion of the MultiClass ICR, the class means on the embeddings $v_1, \ldots, v_k$ are all orthogonal (up to several digits of precision). To complete the definition of the function $f_2$, we add a final transformation step that aligns $v_1, \ldots, v_k$ to the first $k$ coordinate axes. This step is defined by a single orthogonal matrix, so it does not change Euclidean distance or dot products.

**Algorithm 4** BinaryDCR($X, X_1, X_2$)

---
1: $v_1, v_2 \leftarrow$ normalized means($X_1, X_2$)
2: $\theta' \leftarrow$ angle between $v_1$ and $v_2$; $\theta = \frac{\pi}{2} - \theta'$
3: **if** $(\theta' \leq \frac{\pi}{2})$ **then** set angle $\phi = \theta'/2$
           **else** set angle $\phi = \frac{\pi}{4}$
4: **for** $x \in \{x \in X \mid \langle v_2, x \rangle \leq \phi\}$ **do**
5:     $x \leftarrow R_\theta x$

---

Discontinuous Class Rectification (DCR): This approach is similar but does not require iteration at the expense of a discontinuous operation. It replaces the graded rotation (Dev et al., 2021) with a step that identifies a conical region around $v_2$ and applies an angle $\phi$ to all points in this region so afterward $\langle v_1, v_2 \rangle = 0$. If the angle between $v_1$ and $v_2$ is acute, then the conical region is defined in the span of $v_1, v_2$ by an angle $\theta$ from $v_2$ to the bisector direction between $v_1$ and $v_2$. That is, for points closer to $v_2$, they are moved along with $v_2$, and the rest are left alone. If $v_1$ and $v_2$ have an obtuse angle, then we set the conical angle around $v_2$ at $\pi/4$, so we only move points which will be closer to $v_2$ *after* the transformation when $\langle v_1, v_2 \rangle = 0$. The multiclass version of DCR follows the Gram-Schmidt recipe of ICR but with no iteration.

**Freezing learned embedding $X'$.** It is important to note that before ICR or DCR is applied to determine $X$, we need to learn and *freeze* the initial embedding $X' \leftarrow f_1(Z)$. Then $f_2$ operates on $X'$, to create $X \leftarrow f_2(X')$ without adjusting $f_1$. There are slight differences in how OPL (Ranasinghe et al., 2021) and CIDER (Ming et al., 2023) choose an embedding layer: for OPL, it is the penultimate layer, whereas for CIDER, it is the "head," the last layer. We follow recommendations in those works.

In evaluation mode, we also need a classifier. In Section 3.2, we discuss two ways to train classifiers – one is the Rocchio classifier (which we recommend for its structural properties and since it needs no further training). However, a common approach is to build a logistic regression model on the last layer of $f_2(f_1(Z))$; we also do this on the training data. Finally, we can consider the evaluation/test data $z \in Z_{\text{test}}$, which are evaluated with the chosen classifier operating on $f_2(f_1(z))$.

## 2.1 Properties of ICR and DCR

We highlight key properties of the ICR and DCR procedures. Technical proofs are deferred to Appendix B.

First, we show that binary ICR, even if iterated, only affects the coordinates of data points in the span of the original mean vectors. This implies that the mean vectors of classes stay in their original span. Moreover, it implies that as MultiICR gradually includes more classes, it maintains a modified span, and all components of coordinates outside those spans are unchanged. Hence, if $d > k$, then the null space of the classes is unchanged under the MultiICR procedure. These results follow trivially for binaryDCR and MultiDCR since we just apply the Gram-Schmidt procedure to class cones (the cones around the class that contain the whole associated class).

Second, we show that this process converges to have the mean vectors completely orthogonal to each other. This argument requires that initial classes $X'_j$ are clustered; this explains and justifies the use of optimizing $f_1$ under the OPL or CIDER loss, or something similar, before applying BinaryICR. The assumption we use (Assumption 1; see also Appendix B.1) is probably more restrictive than necessary (it requires clusters to be completely separable), but it makes already messy proofs manageable.

**Assumption 1** *Let $v_i$ be the mean of $X_i$, and let $X_i$ be included in the cone of radius $\phi_i$ around $v_i$ for $i = 1, 2, \ldots, k$. Assume these cones are disjoint (except at the origin). Figure 4 illustrates $k = 2$ and $k = 3$.*

**Theorem 1 (Convergence of BinaryICR)** *Let Assumption 1 hold with $k = 2$, and the angle between $v_1$ and $v_2$ is less than $\frac{\pi}{2}$. Then the BinaryICR algorithm converges: as we iterate, in the limit, the angle between class means approaches $\frac{\pi}{2}$.*

*Proof Sketch.* We consider the positive angle $\gamma$, the gap between the upper bound of the cones of radius $\phi_1$ and the lower bound of the cone with radius $\phi_2$ (see Figure 4). We then prove that after each iteration of BinaryICR, the angle between the new means of two classes does not exceed $\frac{\pi}{2}$ (see Lemma 4). This helps to show that after each iteration of BinaryICR, the gap $\gamma$ increases. Therefore, we end up with an increasing sequence of positive real numbers bounded by $\frac{\pi}{2}$, which is convergent. Lastly, we discuss that the convergence is to $\frac{\pi}{2}$, showing the means of two classes after convergence are orthogonal to each other. □

The comparable arguments for DCR are more straightforward. Following Assumption 1, all points of $X'_2$ are in a cone, and all of them and only them are updated in the operation. Since those points are all moved at an angle exactly $\phi$, and $\phi$ moves $v_2$ orthogonal to $v_1$, then if we recalculate $v_2$ after the move, it will still be

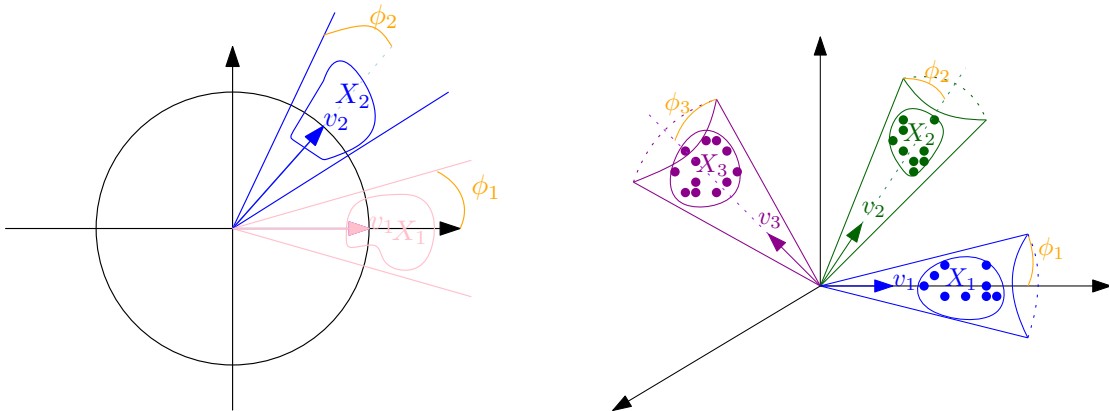

Figure 4: Pictorial view of Assumption 1. Left: $k = 2$, Right: $k = 3$.

orthogonal to $v_1$. Hence, this achieves the orthogonality goal after one step and only affects data in the span of $v_1, v_2$.

**Theorem 2 (Convergence of BinaryDCR)** *Assume Assumption 1 holds with $k = 2$. In addition, if the angle between $v_1$ and $v_2$ is bigger than $\frac{\pi}{2}$, then we assume $\phi_1, \phi_2$ are less than $\frac{\pi}{4}$. Then, after running the BinaryDCR algorithm, the class means will be orthogonal to each other.*

*Proof Sketch.* We basically run the Gram-Schmidt algorithm on the two class cones instead of class means. Under Assumption 3, the whole class $X_2$ will be rotated by a fixed angle, the gap between $v_2$ and $y$-axis. Thus the class mean $v_2$ will be located on $y$-axis. As the first class mean $v_1$ is supposed to be on the $x$-axis, the two class means are now completely orthogonal to each other. $\square$

However, data may not be completely separable; we experimentally observe that OPL and CIDER achieve 99-100% We observed that the difference in output from the original and robust version is in the third digit of precision, so we only report results for the simpler non-robust variant of DCR.

The MultiDCR algorithm is running the Gram-Schmidt algorithm on class cones such that normalized class means will constitute an orthonormal basis for a $k$-dimensional subspace of $\mathbb{R}^d$.

**Theorem 3 (Convergence of MultiDCR)** *Let Assumption 1 hold. In addition, suppose that cones are sufficiently well-separated (see Assumption 3 in Appendix B.2). Then, after running the MultiDCR algorithm, all class means will be orthogonal to each other.*

## 3 Experiments

We evaluate our methods, ICR and DCR, in two main ways. First, we show that these approaches, with high precision, achieve orthogonality of class means while previous approaches do not and maintain good class compactness. Second, we show these approaches maintain or improve upon the near state-of-the-art accuracy in various learning frameworks. Note that ICR and DCR are designed to *maintain* class cohesiveness, not improve upon it, so we do not expect improvement in training data, and any improvement on the evaluation sets can be seen as a fortuitous effect of regularizing to a meaningful structure. Third, we examine a few example images and how, with OPL or CIDER, they have unwanted associations with other classes, but after applying ICR or DCR, that association mostly disappears.

**Datasets and Training Details.** We use standard image classification data sets, tasks, and basic architectures. In our main experiments, we use Resnet-9 as the backbone architecture for the CIFAR-100 Krizhevsky (2009) classification task and train for 120 epochs. The CIFAR-100 is an image dataset that consists of 60,000 natural images that are distributed across 100 classes with 600 images per class. All

training, including ICR & DCR, is performed on the training samples of 50,000 images. All evaluation is shown on the test data of the remaining 10,000 images.

## 3.1 Orthogonality and Compactness

The dimension of the penultimate layer in OPL (Ranasinghe et al., 2021) that was optimized towards being orthogonal was set to $d = 64$ dimensions. It is mathematically impossible to fit $k$ classes orthogonally for $k > d$ dimensions; note $k = 100$ for CIFAR-100 has $100 = k > d = 64$. Alternatively, CIDER (Ming et al., 2023) uses $d = 512$ dimensions in the final layer where dispersion and compactness are optimized. To help identify the best choice of $d$, we first measure geometric properties for OPL and CIDER for $d = 64, 128, 256, 512, 1024$. Table 1 shows for each: first, the average absolute dot-product between class means $\frac{1}{\binom{k}{2}} \sum_{j \neq j'} |\langle v_j, v_{j'} \rangle|$, and second, the average intra-class compactness $\frac{1}{k} \sum_{j \in [k]} \frac{1}{X_j'} \sum_{x \in X_j'} \langle v_j, x \rangle$. For both, orthogonality increases (average dot products decrease) with higher dimensions, and while OPL's compactness keeps increasing, CIDER's decreases after $d = 128$. Notably, even at $d = 1024$, both OPL and CIDER are still far from orthogonal, with an average dot product of about 0.1.

Table 1: Average class absolute dot products; and intra-class compactness.

| dim: | 64 | 128 | 256 | 512 | 1024 | 64 | 128 | 256 | 512 | 1024 |
|---|---|---|---|---|---|---|---|---|---|---|
| OPL | 0.2709 | 0.2412 | 0.1945 | 0.1509 | 0.1267 | 0.9742 | 0.9784 | 0.9851 | 0.9885 | 0.9900 |
| CIDER | 0.1602 | 0.1435 | 0.1247 | 0.1017 | 0.0930 | 0.9816 | 0.9809 | 0.9789 | 0.9764 | 0.9754 |

Next, in Table 2, we show the top-1 and top-5 accuracy for OPL and CIDER by dimension on the CIFAR-100 evaluation set. OPL performs better than CIDER and has the best top-1 accuracy at 1024 dimensions. Somewhat surprisingly, all others peak at smaller dimensions (128 or 256), but the decrease is mostly not too significant. We decided to continue with the best result for top-1 accuracy and orthogonality, and so set $d = 1024$ dimensions as a default.

In Figure 5, we also plot block matrices for the absolute value of dot products between all pairs of class means for OPL and CIDER embeddings at 64 and 1024 dimensions. While increasing $d$ can be seen to improve orthogonality, none are fully orthogonal. Note CIDER dot products appear more uniform than OPL, but the overall average absolute dot products do not differ much in Table 1.

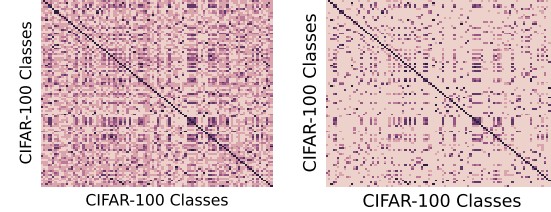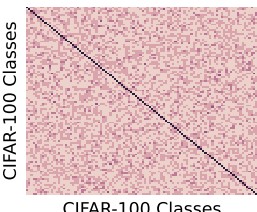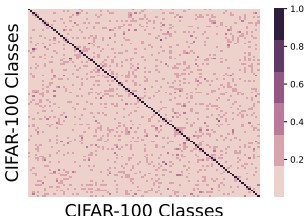

Figure 5: Orthogonality visualization of the dot product of the average per-class feature vectors. From Left to right: OPL(64), OPL(1024), CIDER(64), CIDER(1024).

Thus, OPL and CIDER cannot achieve complete orthogonality of different class features by clustering the same class features. As one of our goals is to translate class indicators to align exactly onto coordinates for interpretability, these loss-function-based approaches are not sufficient.

**Augmentation of OPL features with ICR and DCR.** Next, we add our rectification algorithms, ICR and DCR, on top of the near-orthogonal and compact embeddings as output by OPL or CIDER. We use $d = 1024$ as default but also show the dimensions used in the paper as OPL(64) and CIDER(512). The orthogonality and compactness results are in Table 3. For ICR, we show the result after each of the 5 iterations. Note that ICR improves the average dot product by about one (1) digit of precision in each

Table 2: Softmax Top 1 and Top 5 Accuracy of each d

| Loss | | 64 dim | 128 dim | 256 dim | 512 dim | 1024 dim |
|------|------|--------|---------|---------|---------|----------|
| OPL | Top 1 | 73.38 | 74.29 | 74.26 | 74.87 | 75.22 |
| CIDER | Top 1 | 71.94 | 72.23 | 72.00 | 72.00 | 71.80 |
| OPL | Top 5 | 91.41 | 92.42 | 92.61 | 92.62 | 92.14 |
| CIDER | Top 5 | 89.02 | 89.35 | 89.15 | 89.20 | 88.84 |

iteration, and compactness stays about the same, sometimes increasing. DCR achieves two digits of precision in the average dot product after one step, with a slight degradation in compactness.

Table 3: Orthogonality and Compactness scores for OPL, CIDER, and each after applying +DCR or +ICR $j$, for $j$ iterations. As default, with 1024 dimensions.

| Score | OPL (64) | OPL | OPL+DCR | OPL+ICR 1 | OPL+ICR 2 | OPL+ICR 3 | OPL+ICR 4 | OPL+ICR 5 |
|-------|----------|-----|---------|-----------|-----------|-----------|-----------|-----------|
| Orthogonality | 0.2709 | 0.1268 | 0.0015 | 0.0056 | 0.0006 | $8.2321e\text{-}5$ | $1.1560e\text{-}5$ | $1.7660e\text{-}6$ |
| Compactness | 0.9742 | 0.9899 | 0.9669 | 0.9785 | 0.9779 | 0.9779 | 0.9779 | 0.9779 |

| Score | CIDER (512) | CID | CID+DCR | CID+ICR 1 | CID+ICR 2 | CID+ICR 3 | CID+ICR 4 | CID+ICR 5 |
|-------|-------------|-----|---------|-----------|-----------|-----------|-----------|-----------|
| Orthogonality | 0.1017 | 0.0930 | 0.0057 | 0.0138 | 0.0021 | 0.0004 | $7.4106e\text{-}5$ | $1.5946e\text{-}5$ |
| Compactness | 0.9764 | 0.9754 | 0.9594 | 0.9586 | 0.9566 | 0.9563 | 0.9562 | 0.9562 |

## 3.2 Classification Accuracy after ICR/DCR

We next investigate the effect on classification accuracy after applying ICR and DCR. We now note that there are two standard ways to enact class predictions in this setting. The first is recommended in the OPL paper: build a simple logistic regression for each class and choose the class with the highest score for a query (denoted Smax). In this paper, we prefer using the less powerful model of the Rocchio algorithm $\hat{j} = \arg\max_{j \in [k]} \langle v_j, q \rangle$, for a query $q$ (denoted NN). Table 4 shows the top-1 and top-5 accuracy for OPL, CIDER, and after applying +DCR or +ICR for up to 5 iterations.

Table 4: Test data results for OPL, CIDER and + DCR or +ICR with 1024 dimensions

| Metric | OPL(64) | OPL | OPL+DCR | OPL+ICR 1 | OPL+ICR 2 | OPL+ICR 3 | OPL+ICR 4 | OPL+ICR 5 |
|--------|---------|-----|---------|-----------|-----------|-----------|-----------|-----------|
| Smax Top 1 | 73.20 | 75.28 | 74.47 | 75.21 | 75.19 | 75.19 | 75.20 | 75.20 |
| Smax Top 5 | 91.23 | 91.93 | 89.31 | 91.71 | 91.35 | 91.28 | 91.29 | 91.29 |
| NN Top 1 | 72.36 | 74.57 | 73.39 | 75.02 | 75.03 | 75.03 | 75.03 | 75.03 |
| NN Top 5 | 90.17 | 89.84 | 89.25 | 91.76 | 91.35 | 91.26 | 91.24 | 91.23 |

| Metric | CIDER (512) | CID | CID+DCR | CID+ICR 1 | CID+ICR 2 | CID+ICR 3 | CID+ICR 4 | CID+ICR 5 |
|--------|-------------|-----|---------|-----------|-----------|-----------|-----------|-----------|
| Smax Top 1 | 72.00 | 71.80 | 71.46 | 71.59 | 71.60 | 71.58 | 71.58 | 71.79 |
| Smax Top 5 | 89.20 | 88.84 | 86.02 | 88.26 | 87.72 | 87.60 | 87.60 | 87.67 |
| NN Top 1 | 72.19 | 71.74 | 71.50 | 71.60 | 71.66 | 71.61 | 71.61 | 71.61 |
| NN Top 5 | 89.08 | 88.65 | 85.95 | 88.240 | 87.63 | 87.52 | 87.47 | 87.47 |

For both the Smax (logistic) and NN (Rocchio) classifiers, the OPL initialization outperforms CIDER. Unsurprisingly, the more powerful Smax (logistic) classifier (about 75.2% on top-1) has a bit better performance than the NN (Rocchio) approach (about $74.5 - 75\%$ on top-1). The overall best score is found with just OPL ($d = 1024$) at 75.28% improving upon the baseline OPL ($d = 64$) at 73.20%; applying ICR slightly decreases this to 75.21% or 75.20%. However, for the NN classifier, applying ICR actually improves the score from OPL ($d = 64$) at 72.36% and OPL ($d = 1024$) at 74.57% up to a score of 75.03% – which is not far from

Table 5: OOD performance for CIDER, CIDER+DCR/ICR on CIFAR-100

| | SVHN | | Places365 | | LSUN | | iSUN | | Texture | | **Average** | |
|---|---|---|---|---|---|---|---|---|---|---|---|---|
| | FPR↓ | AUC↑ | FPR↓ | AUC↑ | FPR↓ | AUC↑ | FPR↓ | AUC↑ | FPR↓ | AUC↑ | **FPR↓** | **AUC↑** |
| CE+SimCLR | 24.82 | 94.45 | 86.63 | 71.48 | 56.40 | 89.00 | 66.52 | 83.82 | 63.74 | 82.01 | 59.62 | 84.15 |
| KNN+ | 39.23 | 92.78 | 80.74 | 77.58 | 48.99 | 89.30 | 74.99 | 82.69 | 57.15 | 88.35 | 60.22 | 86.14 |
| OPL | 98.83 | 43.00 | 99.16 | 38.08 | 99.85 | 25.93 | 91.52 | 63.20 | 91.54 | 51.90 | 96.18 | 44.42 |
| CIDER | 44.16 | 89.47 | 69.44 | 80.82 | 57.59 | 86.29 | 9.27 | 98.09 | 35.74 | 91.72 | 43.24 | 89.28 |
| CIDER+DCR | 48.52 | 88.21 | 71.29 | 79.95 | 62.18 | 84.33 | 10.78 | 97.80 | 37.46 | 90.95 | 46.05 | 88.25 |
| CIDER+ICR1 | 49.28 | 87.97 | 70.28 | 79.93 | 60.42 | 84.94 | 10.96 | 97.71 | 37.84 | 91.02 | 45.75 | 88.32 |
| CIDER+ICR2 | 49.72 | 87.92 | 70.53 | 79.89 | 60.51 | 84.86 | 11.08 | 97.70 | 38.03 | 90.99 | 45.97 | 88.27 |

the best Smax (logistic) score. Similar effects are seen with top-5 accuracy (and CIFAR-10 in Appendix C), where OPL outperforms CIDER, and in this case, using ICR has little effect and provides an improvement in NN (Rocchio) classifiers.

To verify that OPL+ICR does not deteriorate representations, we applied it to the training data (see Tables 13 and 14 in Appendix D) where all methods achieve between 99.5% and 100% accuracy; with the exception of some degradation under the Smax (logistic) classifier after using CIDER loss.

### 3.3 Out-of-Distribution Detection

Out-of-Distribution Detection (OOD) is the task of identifying testing samples that originate from an unknown distribution, which the data representation did not encounter during training. This task evaluates the model's dependability when encountering both known in-distribution (ID) inputs and OOD samples – these should not be forced into an existing classification structure and may represent anomalies requiring further attention. A wide variety of OOD detection methods have been explored, with distance-based OOD detection demonstrating considerable potential (Lee et al., 2018; Xing et al., 2019) via representation learning. A central approach extends a Rocchio-type setup and determines ID vs. OOD based on the distance to class means. Very recently, Ming et al. (2023) introduced CIDER, a Compactness and Dispersion Regularized learning framework for OOD detection, discussed earlier in equation 2. This provides a significant improvement in the state of the art.

**Datasets and Training Details** In line with the approach taken by Ming et al. (2023), we adopt the CIFAR-10 and CIFAR-100 (Krizhevsky, 2009) as the in-distribution datasets (CIFAR-10 results in Appendix C). For evaluating the OOD detection performance, we use a diverse collection of natural image datasets encompassing SVHN (Netzer et al., 2011), Places365 (Zhou et al., 2018), Textures (Cimpoi et al., 2013), LSUN (Yu et al., 2015), and iSUN (Xu et al., 2015). Our experiments utilize the pre-trained ResNet-9 used in the Image Classification task for the CIFAR-100 dataset. We freeze the pre-trained model up to the penultimate layer to extract CIDER ID and OOD features for our OOD detection experiments. After obtaining the extracted CIDER features, we apply ICR to refine the features further, enhancing inter-class separation within the feature embedding space. Upon acquiring the ICR-rectified CIDER ID and OOD features at test time, we employ CIDER's distance-based code for OOD detection.

The following metrics are reported in Table 5: 1) False Positive Rate (FPR) of OOD samples at 95% True Positive Rate (TPR) of ID samples, and 2) Area Under the Receiver Operating Characteristic Curve (AUC). We show two representative prior art: CE+SimCLR (Winkens et al., 2020) and KNN+(Sun et al., 2022), the best two methods before CIDER. Observe how CIDER significantly improves FPR from about 60% to about 43% and AUROC from about 84-86% to 89% (averaged across data sets). Applying ICR or DCR shows a small degradation of these improvements, with an FPR of about 45% and AUROC of about 88%, still a significant improvement over the previous baselines, but now with an interpretable structure. On CIFAR-10, CIDER+ICR slightly improves over just CIDER; see Appendix C. This task seems delicate, and for instance, using OPL equation 1 in place of CIDER equation 2 achieves much worse results with an average FPR of 96% and AUROC of only 44%.

### 3.4 Qualitative Example Comparison

In Figure 6 we show a few illustrative examples from CIFAR-100, and compare their predictions on OPL, CIDER, and applying +DCR or +ICR. For each image, we show the top 5 results under the NN (Rocchio) classifier. After ICR or DCR, these are the coordinates in the new coordinate system associated with the $k = 100$ classes.

We observe that while all methods have the label correct as the top prediction, the drop-off after the first prediction is steeper after ICR or DCR is applied. For instance, because under OPL, the class means for "woman" are correlated with other people (e.g., girl, man, boy), under OPL, the woman example also has a high dot-product with those class means. But after ICR or DCR, the class means are orthogonal, so the forced high association is gone. The same phenomenon can be seen with man & boy and with worm & snake.

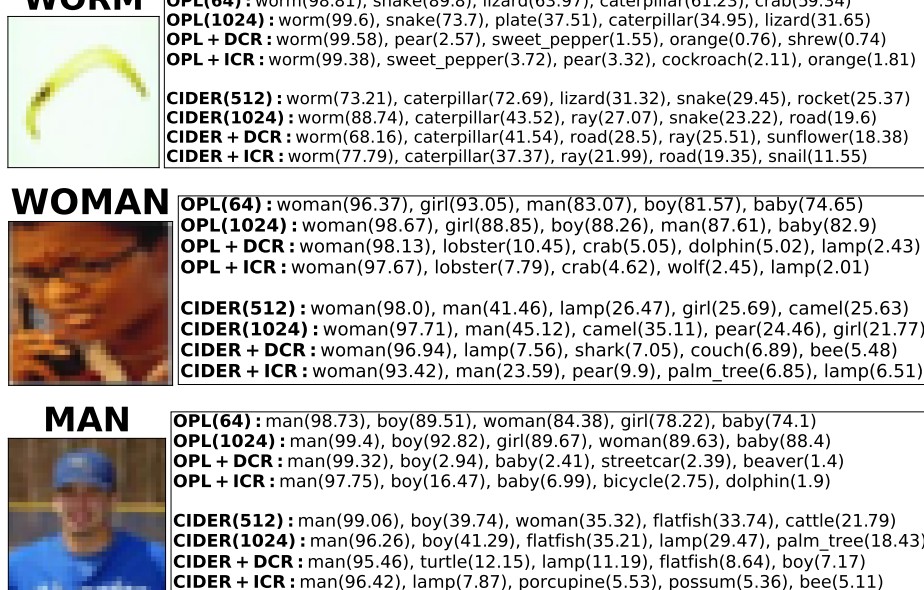

**WORM**
**OPL(64):** worm(98.81), snake(89.8), lizard(63.97), caterpillar(61.23), crab(59.34)
**OPL(1024):** worm(99.6), snake(73.7), plate(37.51), caterpillar(34.95), lizard(31.65)
**OPL + DCR:** worm(99.58), pear(2.57), sweet_pepper(1.55), orange(0.76), shrew(0.74)
**OPL + ICR:** worm(99.38), sweet_pepper(3.72), pear(3.32), cockroach(2.11), orange(1.81)

**CIDER(512):** worm(73.21), caterpillar(72.69), lizard(31.32), snake(29.45), rocket(25.37)
**CIDER(1024):** worm(88.74), caterpillar(43.52), ray(27.07), snake(23.22), road(19.6)
**CIDER + DCR:** worm(68.16), caterpillar(41.54), road(28.5), ray(25.51), sunflower(18.38)
**CIDER + ICR:** worm(77.79), caterpillar(37.37), ray(21.99), road(19.35), snail(11.55)

**WOMAN**
**OPL(64):** woman(96.37), girl(93.05), man(83.07), boy(81.57), baby(74.65)
**OPL(1024):** woman(98.67), girl(88.85), boy(88.26), man(87.61), baby(82.9)
**OPL + DCR:** woman(98.13), lobster(10.45), crab(5.05), dolphin(5.02), lamp(2.43)
**OPL + ICR:** woman(97.67), lobster(7.79), crab(4.62), wolf(2.45), lamp(2.01)

**CIDER(512):** woman(98.0), man(41.46), lamp(26.47), girl(25.69), camel(25.63)
**CIDER(1024):** woman(97.71), man(45.12), camel(35.11), pear(24.46), girl(21.77)
**CIDER + DCR:** woman(96.94), lamp(7.56), shark(7.05), couch(6.89), bee(5.48)
**CIDER + ICR:** woman(93.42), man(23.59), pear(9.9), palm_tree(6.85), lamp(6.51)

**MAN**
**OPL(64):** man(98.73), boy(89.51), woman(84.38), girl(78.22), baby(74.1)
**OPL(1024):** man(99.4), boy(92.82), girl(89.67), woman(89.63), baby(88.4)
**OPL + DCR:** man(99.32), boy(2.94), baby(2.41), streetcar(2.39), beaver(1.4)
**OPL + ICR:** man(97.75), boy(16.47), baby(6.99), bicycle(2.75), dolphin(1.9)

**CIDER(512):** man(99.06), boy(39.74), woman(35.32), flatfish(33.74), cattle(21.79)
**CIDER(1024):** man(96.26), boy(41.29), flatfish(35.21), lamp(29.47), palm_tree(18.43)
**CIDER + DCR:** man(95.46), turtle(12.15), lamp(11.19), flatfish(8.64), boy(7.17)
**CIDER + ICR:** man(96.42), lamp(7.87), porcupine(5.53), possum(5.36), bee(5.11)

Figure 6: Example images and top-5 scoring NN (Rocchio) values among classes in CIFAR-100.

## 4 Conclusion & Discussion

This paper introduces a post-processing to the training phase of a learned embedding mechanism, which provides an interpretable structure. Namely, for a learned embedded representation for a multi-class classification task, our method, Iterative Class Rectification (ICR), continuously adjusts the embedding function so each of $k$ identified class means is associated with a coordinate. Thus, the representation of each class is orthogonal and can be independently measured. This does not preclude an object from having an association with multiple classes, but it decouples those contributions.

This class orthogonality could also be useful if the class is associated with a protected class (e.g., gender, race, etc). By restricting to classifiers that predict labels based on dot products along these class coordinates, we could eliminate association learned about that trait by simply ignoring that coordinate from the representation at the evaluation phase. This pre-processes and makes simple the technique that has become popular in language debiasing (Bolukbasi et al., 2016; Dev & Phillips, 2019; Ravfogel et al., 2020; Wang et al., 2020) which first attempts to identify a linear subspace, and then projects all data in the representation of that subspace.

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

## A  Graded Rotation

Here, we describe the graded rotation algorithm from Dev et al. (2021) for completeness. For an angle $\theta$ we can denote the $2 \times 2$ rotation matrix by $R_\theta$, that is, $R_\theta = \begin{bmatrix} \cos\theta & -\sin\theta \\ \sin\theta & \cos\theta \end{bmatrix}$. The graded rotation of a vector $x$ with respect to the mean vectors $v_1$ and $v_2$ was introduced in Dev et al. (2021), which we recall below.

---

**Algorithm 5** GradedRotat$(v_1, v_2, x)$

---

1: **Input:** Unit vectors $v_1$, $v_2$ in $\mathbb{R}^2$ and $x \in \mathbb{R}^2$
2: Set $\theta' = \arccos(\langle v_1, v_2 \rangle)$ and $\theta = \frac{\pi}{2} - \theta'$
3: Set $\phi_1 = \arccos \langle v_1, \frac{x}{|x|} \rangle$
4: Set $v_2' = v_2 - \langle v_1, v_2 \rangle v_1$
5: Set $d_2 = \arccos \langle v_2', \frac{x}{|x|} \rangle$
6: Compute $\theta_x = \begin{cases} \theta \frac{\phi_1}{\theta'} & \text{if } d_2 > 0 \text{ and } \phi_1 \leq \theta' \\ \theta \frac{\pi - \phi_1}{\pi - \theta'} & \text{if } d_2 > 0 \text{ and } \phi_1 > \theta' \\ -\theta \frac{\pi - \phi_1}{\theta'} & \text{if } d_2 < 0 \text{ and } \phi_1 \geq \pi - \theta' \\ -\theta \frac{\phi_1}{\pi - \theta'} & \text{if } d_2 < 0 \text{ and } \phi_1 < \pi - \theta' \end{cases}$
7: **return** $R_{\theta_x} x$

---

It operates entirely on a data point $x$ that lies in $\mathbb{R}^2$, a span that also contains input vectors $v_1$ and $v_2$. When applied to data in higher dimensions, it is assumed data $x$ has already been projected into this span; that step is not covered here.

It then identifies $v_2$ needs to be rotated (angle $\theta'$) so that it will be orthogonal to $v_1$. Then, based on the angle $\phi_1$ that $x$ makes with $v_1$, it determines how large of a ratio that $x$ should make, denoted $\theta_x$. Note that line 6 shows a case statement. While it is convenient to think about data $x$ that lies in the first quadrant and between $v_1$ and $v_2$, the algorithm should also work for data elsewhere in the span. Depending on whether $\phi_1$ is greater or less than $\theta'$ is one condition. The other condition depends on $d_2$, which determines if $x$ is positive (in the direction of $v_2$ or negative (away from $v_2$, in which case things work symmetrically).

## B  Proofs of convergence of BinaryICR

### B.1  Convergence of BinaryICR and BinaryDCR

In order to prove the convergence of BinaryICR and BinaryDCR algorithms, we need to make the following assumptions on data, which are illustrated in Figure 7. Notice Assumption 1 is a special case of Assumption 2.

**Assumption 2** *Let* $0 < \theta' < \frac{\pi}{2}$, $\theta = \frac{\pi}{2} - \theta'$, $-\phi_1 \leq 0 \leq \phi_2 < \psi_1 \leq \psi_2 \leq \pi$ *and* $\gamma = \psi_1 - \phi_2 > 0$. *Let also* $X_1$ *and* $X_2$ *be subsets of the cones* $\Gamma_1 = \{re^{i\phi} : \phi_1 \leq \phi \leq \phi_2, r > 0\}$ *and* $\Gamma_2 = \{re^{i\psi} : \psi_1 \leq \psi \leq \psi_2, r > 0\}$, *respectively, and* $\theta'$ *be the angle between the mean vectors* $v_1$ *and* $v_2$ *of* $X_1$ *and* $X_2$, *respectively (see Figure 7).*

**Lemma 4** *Under Assumption 2, for any* $i$, *the angle* $\theta_i = \frac{\pi}{2} - \theta_i'$ *stays positive, where* $\theta_1' = \theta'$ *and* $\theta_i'$ *($i \geq 2$) shows the angle between two class means after* $i$-*th iteration of BinaryICR.*

*Proof.*  First, we discuss what happens for the cones $\Gamma_1$ and $\Gamma_2$ after one iteration of BinaryICR. Then, by an induction argument, we conclude that $\theta_i > 0$ for any $i$.

In $\Gamma_1$, the half-cone under $v_1$ shrinks, but the other half expands. It means that the $y$-values of data points in the half-cone under $v_1$ increase, and their $x$-values decrease a bit but stay positive. The same phenomenon happens for the other half of the cone. Since we had an increase in $y$-values, their average will increase as well (i.e., will be positive as it was 0 previously). Therefore, $v_1'$ will be in the first quadrant.

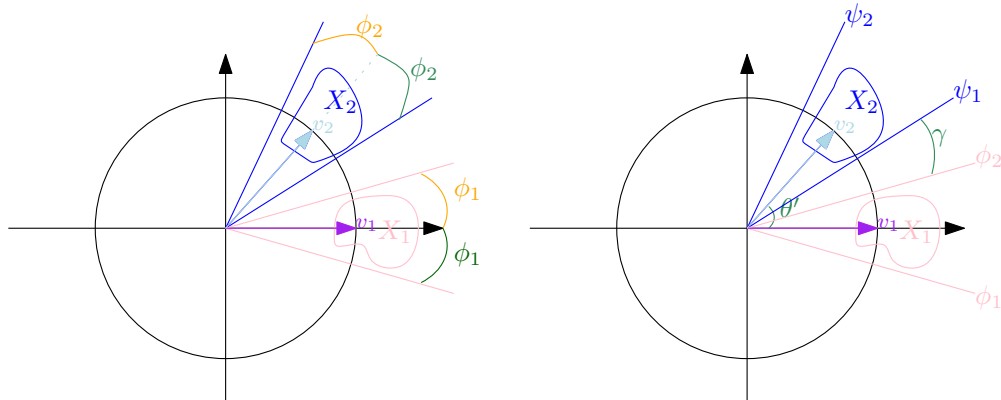

Figure 7: Pictorial view of Assumption 1 (left) and Assumption 2 (right) on two classes $X_1$ and $X_2$.

For $\Gamma_2$, after running one iteration of BinaryICR, the half-cone above $v_2$ shrinks, but the other half expands. Now, in order to make the comparison easy, we rotate all the points of $X_2'$ by $-\theta_1$ (i.e., $y$-axis will be transformed on top of $v_2$) and call it $X_2''$, where $X_2'$ is the transformation of $X_2$ after applying graded rotations on top of $X_2$. This means that the $x$-values of data points in $X_2''$ are increased with respect to their $x$-values in $X_2$ (consider two cases $\psi_2 \leq \pi/2$ and $\psi_2 > \pi/2$ separately). This will also happen to their average, and thus, their average will be under $v_2$. Rotating the points of $X_2''$ back by $\theta_1$ degree to get $X_2'$ means that the average of $X_2'$, which we call $v_2'$, will be less than $\pi/2$.

We observe that both mean vectors $v_1'$ and $v_2'$ stay in the first quadrant, implying $\theta_2' < \pi/2$ or equivalently $\theta_2 > 0$.

Now by an induction argument if $\theta_i > 0$, completely similar to going from $\theta_1 > 0$ to $\theta_2 > 0$ above, we can conclude that $\theta_{i+1} > 0$. $\qquad\square$

**Theorem 5 (Restatement of Theorem 1)** *Under Assumption 2, the BinaryICR algorithm converges, that is, after enough epochs, $\theta_i'$ will approach $\frac{\pi}{2}$, where $\theta_i'$ is the angle between two class means after ith iteration of BinaryICR.*

*Proof.* Let $\theta_1 = \theta$ and $\theta_1' = \theta'$. Notice that all vectors in $(X_1 \cup X_2) \setminus \mathbb{R} \times \{0\}$ will be changed after any iteration of BinaryICR if $\theta_1' \neq \frac{\pi}{2}$. Now consider the gap $\gamma_1 = \gamma$ between $\phi_2$ and $\psi_1$, i.e. $\gamma_1 = \psi_1 - \phi_2 > 0$. Since both $\phi_2$ and $\psi_1$ lie in $[0, \theta_1']$, after one iteration of BinaryICR, they will be mapped to $\phi_2 + \frac{\theta_1}{\theta_1'}\phi_2$ and $\psi_1 + \frac{\theta_1}{\theta_1'}\psi_1$. Thus $\gamma_1$ will be changed to $\gamma_2 = \gamma_1 + \frac{\theta_1}{\theta_1'}\gamma_1 > \gamma_1$ (note $\theta_i \geq 0$ and $0 < \gamma < \theta_i' < \pi/2$ by Lemma 4). Considering $\theta_2'$, running another iteration of BinaryICR will modify $\gamma_2$ to $\gamma_3 = \gamma_2 + \frac{\theta_2}{\theta_2'}\gamma_2 > \gamma_2$ and so on. Therefore, the sequence $(\gamma_n) \subset [0, \frac{\pi}{2}]$ is a bounded increasing sequence and thus convergent, say to $\gamma'$. This means that running another iteration of BinaryICR will not change $\gamma'$, that is $\theta_n \to 0$, otherwise $\gamma'$ will need to be changed. Hence, the BinaryICR algorithm converges. $\qquad\square$

**Theorem 6 (Restatement of Theorem 2)** *Assume Assumption 1 holds. In addition, if the angle between $v_1$ and $v_2$ is bigger than $\frac{\pi}{2}$, then we assume $\phi_1, \phi_2$ are less than $\frac{\pi}{4}$. Then, after running the BinaryDCR algorithm, the class means will be orthogonal to each other.*

*Proof.* The proof is trivial, but we include it for completeness. Let $\theta'$ be the angle between $v_1$ and $v_2$. There are two cases.

*Case 1.* When $\theta' < \frac{\pi}{2}$ and the two classes are disjoint, according to the BinaryDCR algorithm, all vectors in class 2 will be rotated by $\theta = \frac{\pi}{2} - \theta'$ degrees, and so their mean $v_2$ will be rotated by $\theta$ degrees as well. However, the vectors in class 1 will not be rotated. Thus, $v_1$ will stay the same. Therefore, after running the algorithm, the class means will be orthogonal to each other.

*Case 2.* In the case $\theta' > \frac{\pi}{2}$, according to the BinaryDCR algorithm, all the vectors within $\frac{\pi}{4}$ of $v_2$ will be rotated by $\theta = \frac{\pi}{4}$ degrees. So, by the assumptions, all the points in class 2, and thus their mean $v_2$, will be rotated by $\frac{\pi}{4}$ degrees. Since the points in class 1 will stay the same, this means that, after running the algorithm, the class means will be orthogonal to each other. $\qquad\square$

### B.2 Convergence of MultiDCR

**Assumption 3** *We consider the following assumptions on the dataset in order to prove the convergence of the MultiClassDCR algorithm (see Figure 8). Without loss of generality, we assume if we run the Gram-Schmidt process on class means $\{v_1, \ldots, v_k\}$, and it runs successfully (handled by assumption (1)), then the resulting orthonormal basis would be the standard basis $\{e_1, \ldots, e_k\}$.*

1. *Class means are linearly independent.*

2. *For each $i$, class $X_i$ is included in a cone $C_i$ around $v_i$ with radius $\phi_i$, where for $i \geq 3$, $C_i$ is located inside a cone $B_i$ around $e_i$ of radius less than $\pi/2$.*

3. *All class means are in the first orthant, or $\phi_i < \frac{\pi}{4}$ for all $i$.*

4. *For all $j < i$, where $i \geq 3$, $X_j$ is outside of the cone $B_i$.*

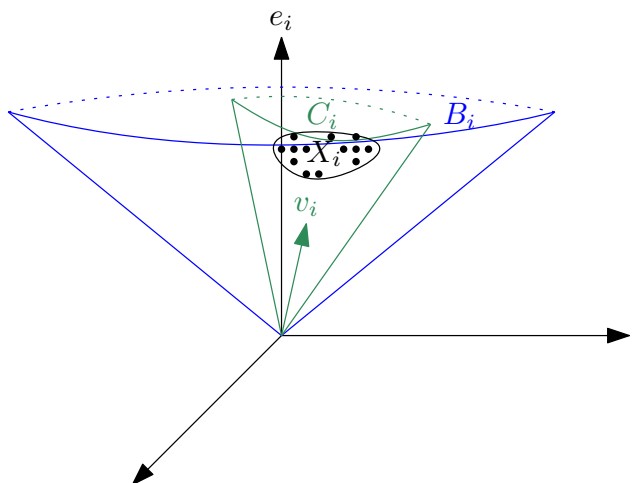

Figure 8: Pictorial view of Assumption 3.

**Theorem 7 (Restatement of Theorem 3)** *Let Assumption 3 hold. Then, after running the MultiDCR algorithm, all class means will be orthogonal to each other.*

*Proof.* In the MultiDCR algorithm, for each class, we rotate the encompassing cone in a Gram-Schmidt manner. Considering the separation assumptions and linear independence property in Assumption 3, all cones will stay separated after any step in the Gram-Schmidt process. This is because, in the Gram-Schmidt process, we orthogonalize vectors one by one; notice that this process happens in the same subspace as before, that is, in the $i$th step the span of $\{e_1, \ldots, e_i\}$ and $\{v_1, \ldots, v_i\}$ will be the same. Now Assumption 3 implies that the $i$th class cone $C_i$ around $v_i$ will be rotated in such a way so that $e_i$ will be its center after the rotation. We call this rotated cone $C_i'$. Thus $C_i'$ will be inside the cone $B_i$, by Assumption 3(2). This means that $C_i'$ will be disjoint from the previously orthogonalized cones $C_j'$ for $j < i$ as they live outside the cone $B_i$ and so outside the cone $C_i'$. Therefore, after running the MultiDCR algorithm, all class means will be orthogonal to each other. $\qquad\square$

## C  Experiments on CIFAR-10

We repeat here on CIFAR-10 many of the experiments we performed for CIFAR-100 in the main text. Things mostly work about the same, but there are some differences due to some accuracy problems being easier due to fewer classes. Also, orthogonality is easier to obtain since there are few classes in the same dimensional space.

### C.1  Orthogonality and Compactness

Like Table 1, Table 6 shows the average absolute dot-product between class means and the average intra-class compactness with OPL and CIDER for CIFAR-10 dataset. For OPL, orthogonality and compactness increase with higher dimensions, while CIDER's orthogonality stays the same and compactness decreases.

Interestingly, CIDAR has worse orthogonality in this case than with CIFAR-100; this is because vectors $v_1, \dots, v_{10}$ must lie in a 10-dimensional span (with the origin), and so one can make them further apart by putting 1/9 of a way into the direction of the origin, hence the 0.111 dot product.

Table 6: Average class dot products;      and intra-class compactness.

| dim: | 64 | 128 | 256 | 512 | 1024 | 64 | 128 | 256 | 512 | 1024 |
|------|----|-----|-----|-----|------|----|-----|-----|-----|------|
| OPL | 0.0111 | 0.0093 | 0.0083 | 0.0036 | 0.0058 | 0.9989 | 0.9990 | 0.9990 | 0.9990 | 0.9991 |
| CIDER | 0.1111 | 0.1111 | 0.1111 | 0.1111 | 0.1111 | 0.9892 | 0.9885 | 0.9880 | 0.9875 | 0.9859 |

Table 7, as Table 4, shows the top-1 and top-5 accuracy for OPL, CIDER, and after applying +DCR or +ICR for up to 5 iterations for CIFAR-10 dataset. Here, as noted in Section 3.2, Smax means applying logistic regression and choosing the class with the highest score for a query, and NN stands for applying the Rocchio algorithm to infer the class predictions.

Table 7: Softmax Top 1 and Top 5 Accuracy of each k

| Loss | | 64 dim | 128 dim | 256 dim | 512 dim | 1024 dim |
|------|-------|--------|---------|---------|---------|----------|
| OPL | Top 1 | 93.020 | 93.610 | 93.200 | 93.420 | 93.310 |
| CIDER | Top 1 | 92.730 | 92.640 | 92.870 | 92.730 | 92.590 |
| OPL | Top 5 | 99.590 | 99.570 | 99.610 | 99.570 | 99.650 |
| CIDER | Top 5 | 99.570 | 99.590 | 99.550 | 99.520 | 99.690 |

### C.2  Orthogonality visualization

In Figure 9, we plot block matrices for the absolute value of dot products between all pairs of class means for OPL and CIDER embeddings at 64 and 1024 dimensions for the CIFAR-10 dataset. For OPL, all classes but two are almost orthogonal in 64 dimensions and this trend is improved in 1024 dimensions, but those two classes are not orthogonal to each other yet. For CIDER, going from 64 to 1024 dimensions does not considerably improve class orthogonality scores and is far from orthogonality.

### C.3  Augmentation of OPL features with ICR and DCR

Table 8 below shows the average absolute dot-product between class means and the average intra-class compactness for the CIFAR-10 dataset when we apply ICR/DCR on top of the OPL/CIDER features. Note that ICR and DCR improve dot products drastically for both OPL and CIDER, where OPL+ICR5 reaches complete orthogonality of classes, and CIDER+ICR5 improves the average dot product by about one (1) digit. Compactness stays about the same for OPL but decreases for CIDER as we iterate more ICR steps.

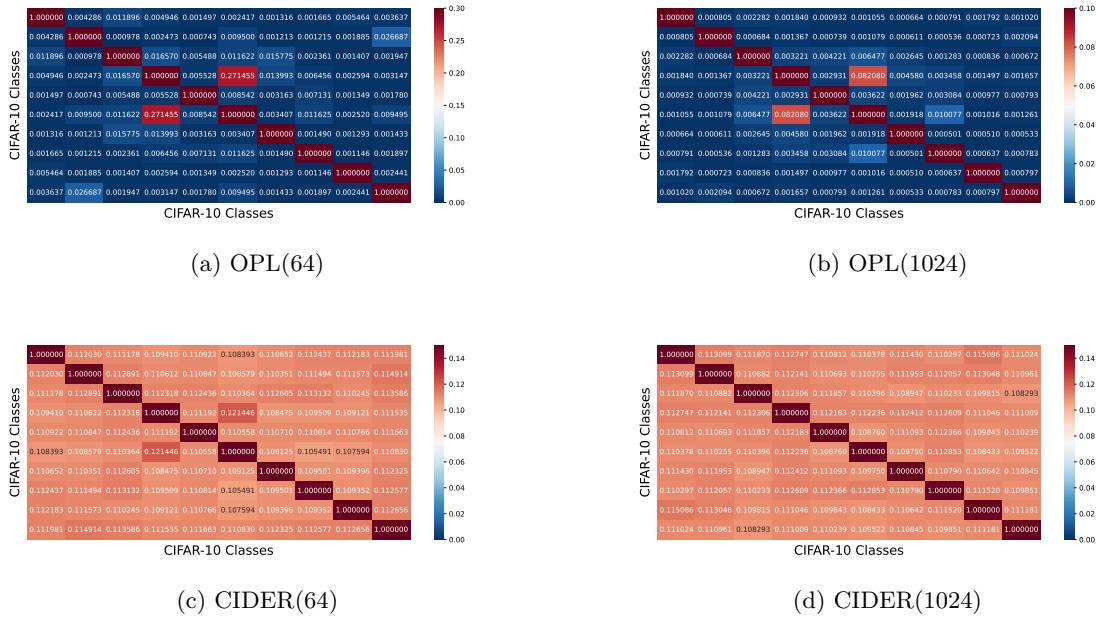

Figure 9: Orthogonality visualization of the dot product of the average per-class feature vectors. From Left to right: OPL(64), OPL(1024), CIDER(64), CIDER(1024).

Table 8: Orthogonality and Compactness scores for OPL, CIDER, and each after applying +DCR or +OPL $j$, for $j$ iterations. As default, with 1024 dimensions.

| Score | OPL (64) | OPL | OPL+DCR | OPL+ICR 1 | OPL+ICR 2 | OPL+ICR 3 | OPL+ICR 4 | OPL+ICR 5 |
|---|---|---|---|---|---|---|---|---|
| Orthogonality | 0.0111 | 0.0058 | $2.0720e\text{-}05$ | $5.2844e\text{-}05$ | $1.0261e\text{-}06$ | $1.8735e\text{-}08$ | $3.2816e\text{-}10$ | $5.7593e\text{-}12$ |
| Compactness | 0.9989 | 0.9991 | 0.9991 | 0.9991 | 0.9991 | 0.9991 | 0.9991 | 0.9991 |

| Score | CIDER (512) | CID | CID+DCR | CID+ICR 1 | CID+ICR 2 | CID+ICR 3 | CID+ICR 4 | CID+ICR 5 |
|---|---|---|---|---|---|---|---|---|
| Orthogonality | 0.1111 | 0.1111 | 0.0744 | 0.0838 | 0.0592 | 0.0372 | 0.0239 | 0.0143 |
| Compactness | 0.9875 | 0.9859 | 0.9779 | 0.9351 | 0.8976 | 0.8778 | 0.8662 | 0.8601 |

## C.4 Classification Accuracy after ICR/DCR on Test Data

Similar to Table 4, Table 9 shows the top-1 and top-5 accuracy on CIFAR-10 test data for OPL, CIDER, and after applying +DCR or +ICR for up to 5 iterations, where OPL outperforms CIDER.

For OPL(1024), applying +DCR or +ICR, top-1 accuracy increases a bit with the Smax (logistic) classifier and stays the same with the NN (Rocchio) classifier. In contrast, for CIDER, top-1 accuracy decreases by an insignificant amount with both Smax and NN classifiers. For both OPL and CIDER, top-5 accuracy degrades by less than 1%.

Table 9: Test data results for OPL, CIDER and + DCR or +ICR with 1024 dimensions

| Metric | OPL(64) | OPL | OPL+DCR | OPL+ICR 1 | OPL+ICR 2 | OPL+ICR 3 | OPL+ICR 4 | OPL+ICR 5 |
|---|---|---|---|---|---|---|---|---|
| Smax Top 1 | 93.020 | 93.310 | 93.330 | 93.330 | 93.330 | 93.330 | 93.330 | 93.330 |
| Smax Top 5 | 99.590 | 99.650 | 98.700 | 98.700 | 98.700 | 98.700 | 98.700 | 98.700 |
| NN Top 1 | 93.030 | 93.300 | 93.290 | 93.300 | 93.300 | 93.300 | 93.300 | 93.300 |
| NN Top 5 | 99.560 | 99.720 | 98.900 | 98.920 | 98.920 | 98.920 | 98.920 | 98.920 |
| Metric | CIDER (512) | CID | CID+DCR | CID+ICR 1 | CID+ICR 2 | CID+ICR 3 | CID+ICR 4 | CID+ICR 5 |
| Smax Top 1 | 92.730 | 92.590 | 92.360 | 92.630 | 92.610 | 92.440 | 92.370 | 92.400 |
| Smax Top 5 | 99.520 | 99.690 | 99.180 | 99.630 | 99.610 | 99.580 | 99.570 | 99.590 |
| NN Top 1 | 92.730 | 92.560 | 92.180 | 92.420 | 91.210 | 90.000 | 89.940 | 89.940 |
| NN Top 5 | 99.420 | 99.550 | 99.010 | 99.170 | 98.870 | 96.840 | 95.510 | 95.350 |

## C.5 Classification Accuracy after ICR/DCR on Training Data

Tables 10 and 11 show the top-1 and top-5 accuracy for OPL, CIDER, and after applying +DCR or +ICR on training data. Similar to CIFAR-100, we observe that OPL+ICR does not deteriorate representations, where all methods achieve between 99.9% and 100% accuracy. Applying +DCR after CIDER affects the accuracy by less than 1%. However, we see some degradation after applying +ICR on top of CIDER features, especially with the Smax classifier, when we use more iterations.

Table 10: Training data results for OPL, OPL+DCR, and OPL+ICR with 1024 dimensions

| Metric | OPL (64) | OPL (1024) | OPL+DCR | OPL+ICR1 | OPL+ICR2 | OPL+ICR3 | OPL+ICR4 | OPL+ICR5 |
|---|---|---|---|---|---|---|---|---|
| Smax Top 1 | 99.976 | 99.976 | 99.976 | 99.976 | 99.976 | 99.976 | 99.976 | 99.976 |
| Smax Top 5 | 99.998 | 100.000 | 100.000 | 100.000 | 100.000 | 100.000 | 100.000 | 100.000 |
| NN Top 1 | 99.974 | 93.300 | 93.290 | 99.974 | 99.974 | 99.974 | 99.974 | 99.974 |
| NN Top 5 | 100.000 | 99.720 | 98.900 | 100.000 | 100.000 | 100.000 | 100.000 | 100.000 |

Table 11: Training data results for CIDER, CIDER+DCR, and CIDER+ICR with 1024 dimensions

| Metric | CIDER (512) | CID | CID+DCR | CID+ICR 1 | CID+ICR 2 | CID+ICR 3 | CID+ICR 4 | CID+ICR 5 |
|---|---|---|---|---|---|---|---|---|
| Smax Top 1 | 99.944 | 99.952 | 99.150 | 94.176 | 93.402 | 87.858 | 86.980 | 86.890 |
| Smax Top 5 | 100.000 | 100.000 | 99.998 | 98.456 | 93.588 | 93.582 | 93.192 | 87.336 |
| NN Top 1 | 99.946 | 99.950 | 99.716 | 99.872 | 98.178 | 96.554 | 96.478 | 96.474 |
| NN Top 5 | 100.000 | 100.000 | 100.000 | 100.000 | 99.926 | 97.570 | 96.590 | 96.572 |

## C.6 Full Table for Out of Distribution Experiment using CIFAR-10 as In-Distribution (ID) Data

We show results in Table 15 for CIFAR-10 on the OOD experiments for which CIDER gave state-of-the-art results. Many prior art results are taken directly from Ming et al. (2023). Unlike in CIFAR-100, where running ICR gives minor degradation of results under these measures, with CIFAR-10, ICR and DCR give barely measurable improvement or degradation (in the fourth bit of precision). OPL still does not perform as well on this task.

Table 12: OOD performance for for CIDER, CIDER+DCR, and CIDER+ICR on the CIFAR10 Dataset

| Method | OOD Dataset | | | | | | | | | | | |
| | SVHN | | Places365 | | LSUN | | iSUN | | Texture | | **Average** | |
| | **FPR↓** | **AUC↑** | **FPR↓** | **AUC↑** | **FPR↓** | **AUC↑** | **FPR↓** | **AUC↑** | **FPR↓** | **AUC↑** | **FPR↓** | **AUC↑** |
|---|---|---|---|---|---|---|---|---|---|---|---|---|
| **Without Contrastive Learning** | | | | | | | | | | | | |
| MSP | 59.66 | 91.25 | 62.46 | 88.64 | 45.21 | 93.80 | 54.57 | 92.12 | 66.45 | 88.50 | 57.67 | 90.86 |
| ODIN | 53.78 | 91.30 | 43.40 | 90.98 | 10.93 | 97.93 | 28.44 | 95.51 | 55.59 | 89.47 | 38.43 | 93.04 |
| Mahalanobis | 9.24 | 97.80 | 83.50 | 69.56 | 67.73 | 73.61 | 6.02 | 98.63 | 23.21 | 92.91 | 37.94 | 86.50 |
| Energy | 54.41 | 91.22 | 42.77 | 91.02 | 10.19 | 98.05 | 27.52 | 95.59 | 55.23 | 89.37 | 38.02 | 93.05 |
| GODIN | 18.72 | 96.10 | 55.25 | 85.50 | 11.52 | 97.12 | 30.02 | 94.02 | 33.58 | 92.20 | 29.82 | 92.97 |
| **With Contrastive Learning** | | | | | | | | | | | | |
| ProxyAnchor | 39.27 | 94.55 | 43.46 | 92.06 | 21.04 | 97.02 | 23.53 | 96.56 | 42.70 | 93.16 | 34.00 | 94.67 |
| CE+SimCLR | 6.98 | 99.22 | 54.39 | 86.70 | 64.53 | 85.60 | 59.62 | 86.78 | 16.77 | 96.56 | 40.46 | 90.97 |
| CSI | 37.38 | 94.69 | 38.31 | 93.04 | 10.63 | 97.93 | 10.36 | 98.01 | 28.85 | 94.87 | 25.11 | 95.71 |
| SSD+ | 2.47 | 99.51 | 22.05 | 95.57 | 10.56 | 97.83 | 28.44 | 95.67 | 9.27 | 98.35 | 14.56 | 97.38 |
| KNN+ | 2.70 | 99.61 | 23.05 | 94.88 | 7.89 | 98.01 | 24.56 | 96.21 | 10.11 | 97.43 | 13.66 | 97.22 |
| **Regularization for Compactness and Dispersion** | | | | | | | | | | | | |
| CIDER | 8.30 | 98.46 | 21.37 | 95.93 | 9.63 | 98.18 | 0.68 | 99.79 | 27.75 | 94.45 | 13.55 | 97.36 |
| CIDER+DCR | 8.33 | 98.46 | 21.58 | 95.92 | 9.59 | 98.19 | 0.67 | 99.79 | 27.75 | 94.48 | 13.58 | 97.37 |
| CIDER+ICR1 | 8.31 | 98.46 | 21.33 | 95.93 | 9.48 | 98.19 | 0.69 | 99.79 | 27.82 | 94.44 | 13.53 | 97.36 |
| CIDER+ICR2 | 8.32 | 98.46 | 21.29 | 95.93 | 9.46 | 98.19 | 0.69 | 99.79 | 27.84 | 94.45 | 13.52 | 97.36 |
| CIDER+ICR3 | 8.32 | 98.46 | 21.30 | 95.93 | 9.46 | 98.19 | 0.69 | 99.79 | 27.84 | 94.45 | 13.52 | 97.36 |
| CIDER+ICR4 | 8.32 | 98.46 | 21.29 | 95.93 | 9.46 | 98.19 | 0.69 | 99.79 | 27.84 | 94.45 | 13.52 | 97.36 |
| CIDER+ICR5 | 8.32 | 98.46 | 21.29 | 95.93 | 9.46 | 98.19 | 0.69 | 99.79 | 27.84 | 94.45 | 13.52 | 97.36 |
| OPL | 99.74 | 33.74 | 99.44 | 42.56 | 99.75 | 56.45 | 99.93 | 49.33 | 97.15 | 40.49 | 99.20 | 44.51 |

## D  Training Data Experiments on Accuracy for CIFAR-100

Tables 13 and 14 show the top-1 and top-5 accuracy for OPL, CIDER, and after applying +DCR or +ICR on training data of CIFAR-100 dataset. Table 13 confirms that OPL+ICR does not deteriorate representations, where all methods achieve between 99.5% and 100% accuracy. In CIDER, with the NN classifier, accuracies remain the same (above 99.7%), but with the Smax classifier, we see some degradation after applying +DCR or +ICR.

Table 13: Training data results for OPL, OPL+DCR, and OPL+ICR with 1024 dimensions

| Metric | OPL (64) | OPL (1024) | OPL+DCR | OPL+ICR1 | OPL+ICR2 | OPL+ICR3 | OPL+ICR4 | OPL+ICR5 |
|---|---|---|---|---|---|---|---|---|
| Smax Top 1 | 99.946 | 99.976 | 99.762 | 99.594 | 99.686 | 99.698 | 99.698 | 99.698 |
| Smax Top 5 | 100.000 | 100.000 | 99.992 | 100.000 | 100.000 | 100.000 | 100.000 | 100.000 |
| NN Top 1 | 99.858 | 99.972 | 99.222 | 99.974 | 99.974 | 99.974 | 99.974 | 99.974 |
| NN Top 5 | 100.000 | 100.000 | 100.000 | 100.000 | 100.000 | 100.000 | 100.000 | 100.000 |

Table 14: Training data results for CIDER, CIDER+DCR, and CIDER+ICR with 1024 dimensions

| Metric | CIDER (512) | CID | CID+DCR | CID+ICR 1 | CID+ICR 2 | CID+ICR 3 | CID+ICR 4 | CID+ICR 5 |
|---|---|---|---|---|---|---|---|---|
| Smax Top 1 | 99.888 | 99.892 | 94.370 | 97.396 | 84.540 | 82.798 | 82.612 | 82.572 |
| Smax Top 5 | 100.000 | 100.000 | 98.266 | 99.408 | 89.178 | 88.088 | 87.920 | 87.902 |
| NN Top 1 | 99.890 | 99.898 | 99.720 | 99.872 | 99.864 | 99.862 | 99.862 | 99.862 |
| NN Top 5 | 100.000 | 100.000 | 99.988 | 100.000 | 99.998 | 99.998 | 99.998 | 99.998 |

# E   Full Table for Out of Distribution Experiment using CIFAR-100 as In-Distribution (ID) Data

This table is a full version of Table 5, where we have added some methods without contrastive learning and a few more iterations of ICR on CIDER features. The numbers for additional methods are pulled from Ming et al. (2023).

Table 15: OOD performance for for CIDER, CIDER+DCR, and CIDER+ICR on the CIFAR10 Dataset

| Method | OOD Dataset | | | | | | | | | | | |
| | SVHN | | Places365 | | LSUN | | iSUN | | Texture | | **Average** | |
| | **FPR↓** | **AUC↑** | **FPR↓** | **AUC↑** | **FPR↓** | **AUC↑** | **FPR↓** | **AUC↑** | **FPR↓** | **AUC↑** | **FPR↓** | **AUC↑** |
| **Without Contrastive Learning** | | | | | | | | | | | | |
| MSP | 78.89 | 79.8 | 84.38 | 74.21 | 83.47 | 75.28 | 84.61 | 74.51 | 86.51 | 72.53 | 83.12 | 75.27 |
| ODIN | 70.16 | 84.88 | 82.16 | 75.19 | 76.36 | 80.1 | 79.54 | 79.16 | 85.28 | 75.23 | 78.7 | 79.11 |
| Mahalanobis | 87.09 | 80.62 | 84.63 | 73.89 | 84.15 | 79.43 | 83.18 | 78.83 | 61.72 | 84.87 | 80.15 | 79.53 |
| Energy | 66.91 | 85.25 | 81.41 | 76.37 | 59.77 | 86.69 | 66.52 | 84.49 | 79.01 | 79.96 | 70.72 | 82.55 |
| GODIN | 74.64 | 84.03 | 89.13 | 68.96 | 93.33 | 67.22 | 94.25 | 65.26 | 86.52 | 69.39 | 87.57 | 70.97 |
| LogitNorm | 59.6 | 90.74 | 80.25 | 78.58 | 81.07 | 82.99 | 84.19 | 80.77 | 86.64 | 75.6 | 78.35 | 81.74 |
| **With Contrastive Learning** | | | | | | | | | | | | |
| ProxyAnchor | 87.21 | 82.43 | 70.1 | 79.84 | 37.19 | 91.68 | 70.01 | 84.96 | 65.64 | 84.99 | 66.03 | 84.78 |
| CE+SimCLR | 24.82 | 94.45 | 86.63 | 71.48 | 56.4 | 89 | 66.52 | 83.82 | 63.74 | 82.01 | 59.62 | 84.15 |
| CSI | 44.53 | 92.65 | 79.08 | 76.27 | 75.58 | 83.78 | 76.62 | 84.98 | 61.61 | 86.47 | 67.48 | 84.83 |
| SSD+ | 31.19 | 94.19 | 77.74 | 79.9 | 79.39 | 85.18 | 80.85 | 84.08 | 66.63 | 86.18 | 67.16 | 85.9 |
| KNN+ | 39.23 | 92.78 | 80.74 | 77.58 | 48.99 | 89.3 | 74.99 | 82.69 | 57.15 | 88.35 | 60.22 | 86.14 |
| **Regularization for Compactness and Dispersion** | | | | | | | | | | | | |
| CIDER | 44.16 | 89.47 | 69.44 | 80.82 | 57.59 | 86.29 | 9.27 | 98.09 | 35.74 | 91.72 | 43.24 | 89.28 |
| CIDER+DCR | 48.52 | 88.21 | 71.29 | 79.95 | 62.18 | 84.33 | 10.78 | 97.8 | 37.46 | 90.95 | 46.05 | 88.25 |
| CIDER+ICR 1 | 49.28 | 87.97 | 70.28 | 79.93 | 60.42 | 84.94 | 10.96 | 97.71 | 37.84 | 91.02 | 45.75 | 88.32 |
| CIDER+ICR 2 | 49.72 | 87.92 | 70.53 | 79.89 | 60.51 | 84.86 | 11.08 | 97.7 | 38.03 | 90.99 | 45.97 | 88.27 |
| CIDER+ICR 3 | 49.82 | 87.9 | 70.6 | 79.88 | 60.59 | 84.84 | 11.15 | 97.7 | 38.07 | 90.98 | 46.05 | 88.26 |
| CIDER+ICR 4 | 49.85 | 87.9 | 70.61 | 79.87 | 60.62 | 84.84 | 11.15 | 97.7 | 38.16 | 90.98 | 46.08 | 88.26 |
| CIDER+ICR 5 | 49.84 | 87.9 | 70.57 | 79.87 | 60.59 | 84.84 | 11.15 | 97.7 | 38.12 | 90.98 | 46.05 | 88.26 |
| OPL | 98.83 | 43 | 99.16 | 38.08 | 99.85 | 25.93 | 91.52 | 63.2 | 91.54 | 51.9 | 96.18 | 44.42 |

