# OpenReview forum: "One-Hot Encoding Strikes Back: Fully Orthogonal Coordinate-Aligned Class Representations"
_TMLR — Rejected by TMLR_

### Review · Reviewer_TiZv · 2024-03-01

**Summary Of Contributions:**

The paper proposes two methods called Iterative Class Rectification (ICR) and Discontinuous Class Rectification (DCR) that projects the learned representations of examples, such as images, so that their class means become orthogonal. The authors argue that orthogonality enables interpretability, since with an appropriate rotation, each class can correspond to a single coordinate. They also mention that this approach doesn't hurt classification results and can help in debiasing models; e.g. by leaving out the coordinate that corresponds to a particular sensitive attribute such as gender.

**Audience:**

No

**Broader Impact Concerns:**

I believe that the claim about using this method for debiasing data is unfounded, and is reminiscent of "fairness through unawareness" that is, by now, well-known that it doesn't work. See the reference above and its related works.

**Claims And Evidence:**

No

**Requested Changes:**

- The paper should clearly motivate why orthogonality in the class means matters. I understand that the authors have tried to motivate this in several places in the paper but I don't think those are valid motivations. As I mentioned above, what the authors refer to as "interpretability" can be simply achieved by representing an example with class probability scores. The claim it helps in debiasing is invalid. And, the argument that the model will not confuse classes is not well-justified (see for example my comment above about Figure 6).

- The experiments are done on two small datasets only and one small model, and the evidence is not convincing as I mention above. DCR does hurt performance in both datasets. Adding more datasets and models would strengthen the paper.

- Please fix the typos and remove the WLOG clauses in the algorithms. Using a consistent notation would make the algorithms more readable.

**Strengths And Weaknesses:**

*Strengths*
1.  The paper extends a prior algorithm, called ISR, that was used in language for debiasing data. Identifying and extending new applications of existing methods is definitely an advantage.

*Weaknesses*
1. Overall, the primary limitation is in the original motivation. I have honestly failed to see why post-processing the learned representation of a pretrained model such that class means become orthogonal matters. This is especially the case given the clustering/separation assumption made in the paper, which would make it easy to build a linear classifier on top of the representations. The authors argue that orthogonality enables interpretability but this is actually misleading, because what the authors call "interpretability" here boils down to classification. Interpretability is not about how or why the model makes a particular prediction; it is simply about how to infer the class label from the representation. Obviously, the same "interpretability" advantage can be achieved if one simply build a linear classifier on top of the representation, and outputs the probability scores assigned to each class. Each coordinate of those probability scores would correspond to a particular class. Similarly, the claim that one can debias by ignoring/removing the coordinate that corresponds to the sensitive attribute is quite wrong. Suppose, for example, we have an image classification problem and the classes are {"gender", "weight", "height"}. Even if weight and height are *individually* orthogonal to gender (uncorrelated), both *together* can be predictive of gender. Please see the literature on "fairness through unawareness," such as [1] that discusses these pitfalls.

2. The paper claims that ICR and DCR do not impact classification accuracy. But, the empirical results show that they do hurt accuracy! In CIFAR100, for example, DCR reduces the top-1 accuracy of OPL from 75.28% to 74.47% and the top-5 accuracy from 91.93% to 89.31%. It also hurts accuracy in CIDER. ICR may appear to be ok in the main paper but the CIFAR10 results in the appendix show that it also hurts; e.g. reducing top-5 in OPL from 99.65% to 98.7%.

3. * Confusing classes can actually convey useful information. In Figure 6, for example, the baseline model confuses the the image of a "woman" with a "girl". To me, this indicates that it is difficult to infer from the image if it corresponds to a "woman" or a "girl", and that's useful information. After applying the proposed ICR method, however, the model is now confident that it is a "woman" and the next best prediction is a "lobster"! I'm not sure if this is, in fact, a desired outcome.

4. The experiments are limited to two small datasets only: CIFAR100 and CIFAR10, and one model ResNet-9.

5. There are several places where the paper is either imprecise or confusing. For example:
* In Eq 1, $n_1=0$ if we train in the 1-shot setting, which makes the objective function invalid.
* Also in Eq 1, should the absolute sign be inside the summation for $\mathcal{L}_{disp}$?
* In Eq 2, the objective function for dispersion is not minimized by having orthogonal vectors.
* The authors says they "prefer using the less powerful model of the Rocchio algorithm," without supporting that decision. Logistic regression is a simple linear head and works better.
* The values in Table 4 are very close to each other and there no estimate of the error margin. It's not easy to infer if the differences are reliable.
* The OOD experiments are not really OOD. In the OOD setting, once a model is trained, we deploy on new domains/distributions. In CIFAR10, there are OOD variants of the dataset (see for example https://github.com/google/uncertainty-baselines). If my understanding is correct, the authors train on CIFAR and then adjust the post-processing to fit another dataset, such as SVHN.

Some minor comments:
* In Algorithm 3, it would more readable to use the same notation throughout the algorithm. The WLOG clauses are not needed and are confusing. For example, why is it easier to read when we "assume that $t=i$"?
* In the abstract, "from each coordinating". Should this be "from each coordinate"?
* In Section 2, the paper assumes the multiclass setting where $y_i\in[k]$, but Figure 3 assumes a multi-label setting.
* What does "we use image data as an exemplar" mean?

[1] Dwork, Cynthia, Moritz Hardt, Toniann Pitassi, Omer Reingold, and Richard Zemel. "Fairness through awareness," 2012.

---

### Review · Reviewer_JrEE · 2024-03-04

**Summary Of Contributions:**

This paper proposes ICR and DCR algorithms for multi-classification under the Rocchio algorithm, which further orthogonalizes the class means. Convergence analysis to orthogonal representations is provided with proof. The findings are verified by empirical experiments with a comparison with existing works.

**Audience:**

Yes

**Broader Impact Concerns:**

No such concerns.

**Claims And Evidence:**

Yes

**Requested Changes:**

1. The limitations need discussion. From my understanding, the good performance of the proposed method requires Phase I embedding to provide embeddings in several separable clusterings. It is theoretically reflected in Assumption 1, where "cones are disjoint". What if Phase I embedding cannot ensure so? Can you still provide a convergence guarantee theoretically and empirically?

2. I think the method is based on the idea that one class corresponds to one feature since different classes of data are orthogonalized. I feel it is more reasonable if a combination of multiple orthogonal features corresponds to one class.

3. Can the proposed method be applied to solve any tasks beyond classification, e.g., regression tasks?

4. The motivation needs some more clarification. Why do we require "orthogonality"? Any supportive references?

**Strengths And Weaknesses:**

Strengths:
1, Overall, the paper is easy to follow.
2. The logic is clear, and the proposed method makes sense.

Weaknesses:
Please see the requested changes.

---

### Review · Reviewer_7Tz6 · 2024-03-05

**Summary Of Contributions:**

This paper presents a mechanism for postprocessing embedding vectors (learned via some previous method) so that the class-means for each labeled class are close-to-orthogonal. They present two versions of the mechanism, one of which iteratively performs "graded" piecewise rotations along 2D subspaces to bring class means closer to orthogonal, and one which discontinuously makes them orthogonal by modifying all points within given conical regions. The authors compare their approach to previously-proposed regularization methods OPL and CIDER, and show that these previous methods do not reach full orthogonality, whereas the proposed approach gets much closer. They also show that the proposed technique keeps roughly similar classification performance and somewhat improves out-of-distribution detection.

**Audience:**

No

**Broader Impact Concerns:**

No broader impact concerns.

**Claims And Evidence:**

No

**Requested Changes:**

I would like to see the weaknesses W1 and W2 addressed before I would recommend this submission for acceptance.

- If the authors believe their approach has advantages beyond the simple matrix-inversion baseline I describe above, I think they should clearly explain this in the paper and discuss why this simple baseline is insufficient. I also think it would be appropriate to add experimental results comparing against this baseline, to see whether the proposed approaches are actually any better.

- I also think the proofs should also be expanded so that each step is clearly explained, either with formal notation or with a step-by-step diagram that explains what the authors mean by their statements. This seems necessary to meet the bar for correctness.

Beyond this, I think the paper could be improved by better motivating why we want orthogonal class means. In particular, if the model can't distinguish well between different classes, why aren't we losing valuable information by forcing the average representations to be orthogonal anyway? What specifically do you believe this orthogonalized representation will allow us to do (that we couldn't do with the class probabilities or non-orthogonal representations already)? (The paper gestures at implicit bias, but I don't think it goes into this in enough detail to motivate the approach, and it's not obvious to me why forcing class means to be orthogonal would debias the representation instead of just hiding the bias inside the transformation.)

It would also be interesting to discuss connections to neural collapse ([Papyan et al. 2020](https://www.pnas.org/doi/full/10.1073/pnas.2015509117)), which is a phenomenon where, upon reaching near-zero training loss, models end up collapsing their class means to a "simplex equiangular tight frame". This isn't quite the same as being orthogonal but it seems closely related.

**Strengths And Weaknesses:**

## Strengths

### S1. Clearly written

The main paper is clearly written overall and includes useful examples and figures to demonstrate the technique.

### S2. Reasonable experimental results

The experiments show that previously-proposed methods for orthogonalizing representations do not lead to having orthogonal class means, whereas the proposed technique does produce this. Their technique also seems to slightly improve out-of-distribution detection when using a simple distance-from-class-means metric.

## Weaknesses

### W1. The problem considered by this work has a simple closed-form solution

The primary goal of the proposed approach is to find a mapping that adjusts the existing learned embeddings to make the class means orthogonal, and to align them with the standard coordinate axes. The two proposed approaches are both fairly complex, and involve piecewise or discontinuous nonlinear transformations: ICR applies a pairwise graded rotation (a nonlinear operation that interpolates angles) to pairs of dimensions and repeats it until convergence, and DCR directly moves points within certain distances of the class means in a discontinuous fashion.

However, I believe there is a simple closed-form solution to this problem. If our goal is to map each class mean $v_i$ to the standard basis vector $e_i$, we can arrange the class means as columns of a matrix

$$
V = \\begin{bmatrix}
\\vdots & \\vdots &  & \\vdots \\\\
v_1 & v_2 & \cdots & v_n \\\\
\\vdots & \\vdots & & \\vdots \\\\
\\end{bmatrix}
$$

and then define our mapping as $f_2(x) = V^{-1}x$. This immediately ensures that $f_2(v_i) = e_i$, and does not require any nonlinear or discontinuous steps or any iterative procedure. Moreover, the $e_i$ are the new class means, because $f_2$ is linear and thus $f_2(E[X]) = E[f_2(x)]$ by the linearity of expectation.

A bit more care must be taken if the embedding space is not equal to the number of classes, but this seems fairly straightforward as well: Gram-Schmidt orthogonalize the class means in order to identify the subspace spanned by those means, then define the mapping so that vectors in this subspace get transformed with $V^{-1}$, and vectors orthogonal to it remain unchanged; this is still a linear mapping. The authors are already doing a similar thing in their BinaryCR algorithm for a 2D subspace.

It seems to me that this simpler procedure would meet all the stated goals of the proposed approaches in this work. Given that, it seems hard to justify the complexity of the proposed methods, and I think the ICR and DCR approaches described here may not be of interest to the TMLR audience.

### W2. Proofs are informal and not sufficiently detailed

The authors make a number of claims about the convergence of their algorithms. However, the proofs of these claims seem informal and are in my opinion not detailed enough to follow the argument.

As one example, the authors state in their proof of Lemma 4 that "In $Γ_1$, the half-cone under $v_1$ shrinks, but the other half expands. It means that the y-values of data points in the half-cone under $v_1$ increase, and their $x$-values decrease a bit but stay positive. The same phenomenon happens for the other half of the cone." It's not clear to me which half-cones are being referenced here or what it means for them to be "under" a vector, and if one half shrinks and the other expands, what does it mean that "the same phenomenon happens for the other half"?

I similarly had difficulty following the statement "Now, in order to make the comparison easy, we rotate all the points of $X_2$′ by −$θ_1$ (i.e., y-axis will be transformed on top of v ) and call it X′′, where X′ is the transformation of X after applying graded rotations on top of X", and there are more statements like this that follow as well. This is very difficult to follow without either formal statements about what is being compared to what, or at least a diagram showing these rotations and shrinking/expansion.

I also noted that the proof of Theorem 5 doesn't ever explicitly prove that it converges to $\pi/2$ (although I think it may be indirectly implied by the second-to-last statement of the proof).

I did not read the rest of the proofs in detail, since they do not seem clearly written enough to follow and I'm not convinced the complexity of these proofs is justified given the simpler algorithm I described above.

### W3. Somewhat weak motivation

I found the motivation for the approach somewhat weak. It wasn't clear to me what we actually gain from having representations with orthogonal class means, or why simpler solutions wouldn't be sufficient for this.

For instance, if we want to be able to do classification, or look at relative associations between multiple classes, why isn't it enough to look at the post-softmax classification probabilities, given that we are training the model on a labeled dataset to begin with? The softmax probabilities are already axis-aligned, and if the model is confident, those probabilities will already be close to orthogonal. And if the model isn't confident and can't correctly distinguish classes, it's not clear to me why we should postprocess the representations to force them to be orthogonal.

This comes up in the presented example regarding the orange, for instance. It seems reasonable that this image should be associated with other classes, since the model may not be able to say with certainty that this is an orange and not a different kind of fruit. Why is it better to force the class means to be orthogonal, aren't we losing information in that case?

The example with the hamster and the apple makes a bit more sense to me, but I still think the motivation could be explained better. It still seems like it's useful to know if the model can't tell the difference between a hamster and a rabbit or a mouse. But perhaps you're implicitly trying to force the model to be more confident on "typical" examples, so that you can distinguish typical ones from atypical ones (e.g. ones with two classes in the image)?

---

> ### Author Response · Authors · 2024-03-07
>
> Thank you very much for suggesting the closed form approach based on the matrix inverse.  This is a really clean and direct solution the core problem we were studying.  It seems like the "right" solution for $f_2$.
>
> On some reflection, it seems that given this cleaner solution, there is now a substantial amount of content in our paper which is superseded, and not needed.  As such, it seems best that we do not attempt to address all of these issues in this submitted version of the paper.  We will not attempt to respond in detail to all of these points, some of them are no longer relevant.  We agree that this version is not ready for publication.
>
> Thank you to all reviewers and editors for the efficient and informative reviews of our work.

---

### Decision · Action_Editor_GEam · 2024-04-10

**Recommendation:** Reject

**Comment:**

The paper is not ready for publication in its current form, as the authors themselves agreed.

**Audience:**

There is no TMLR audience for the paper in its current form, in part because there is a much simpler solution available to the problem being considered, as one of the reviewers pointed out.

**Claims And Evidence:**

As reviewers pointed out, the proofs in the paper are insufficiently rigorous and thus do not support the claims.